# Role of Parenting Styles in Adolescent Substance Use Cessation: Results from a Brazilian Prospective Study

**DOI:** 10.3390/ijerph16183432

**Published:** 2019-09-16

**Authors:** Mariana Canellas Benchaya, Taís de Campos Moreira, Hilda Maria Rodrigues Moleda Constant, Natália Masiero Pereira, Luana Freese, Maristela Ferigolo, Helena Maria Tannhauser Barros

**Affiliations:** 1Graduate Program in Health Science, Federal University of Health Sciences of Porto Alegre-UFCSPA, Rua Sarmento Leite, 245, Porto Alegre 90050-170, Brazil; taiscmoreira@gmail.com (T.d.C.M.); hildamoleda@gmail.com (H.M.R.M.C.); natymasiero23@gmail.com (N.M.P.); freese@ufcspa.edu.br (L.F.); mari@ufcspa.edu.br (M.F.); helenbar@ufcspa.edu.br (H.M.T.B.); 2Department of Psychology in Lutheran, University of Brazil, ULBRA, Gravataí 94170-240, Brazil; 3Laboratory of Neuropsychopharmacology, Department of Pharmacosciences, Federal University of Health Sciences of Porto Alegre-UFCSPA. Rua Sarmento Leite, 245, Porto Alegre 90050-170, Brazil

**Keywords:** parental styles, teenagers, substance-related disorders

## Abstract

Background: This study aims to identify the association between parenting styles and behavioral changes among adolescents regarding the consumption of alcohol, tobacco, cannabis, cocaine/crack. Methods: A group of ninety-nine adolescents (39 girls and 60 boys), aged 14 to 19 years (17.05 ± 1.51), who called in to a call center that provides counseling to substance users, was followed-up for 30 days. Data collection occurred between March 2009 and October 2015. The adolescents answered questions regarding parental responsiveness and demanding nature on a scale to assess parental styles and provided sociodemographic data, substance abuse consumption characteristics, and the Contemplation Ladder scale score. Results: The parental styles most reported by the adolescents were authoritative (30%) and indulgent (28%). Children who perceived their mothers as having an indulgent style and who had absent fathers presented more difficulties in making behavioral changes to avoid alcohol and cocaine/crack consumption. Conclusion: The study found that parent-child relationships were associated with a lack of change in the adolescent regarding substance use behavior, particularly the consumption of alcohol and cocaine/crack.

## 1. Introduction 

Substance use among adolescents is a worldwide health issue [1]. In Brazil, the magnitude of the occurrence of this consumption among teenagers is mainly shown in epidemiological studies [2,3]. National surveys on the use of drugs by students in 2004 and 2010, carried out by the Brazilian Center for Information on Psychotropic Drugs (CEBRID), showed a significant increase in the use of illicit drugs and a worrisome prevalence at an early age [2]. Moreover, alcohol experimentation is most prevalent among 12- to 14-year-olds [4]. In addition, socioeconomic inequalities in Brazil have been associated with substance use among adolescents, where the higher risk of abuse and dependence are evidenced in less favored socioeconomic classes [1]. These data are worrisome, as early adolescent drug use dramatically increases the risk of lifelong substance use disorder (SUD), interferes with ongoing neurodevelopment, and may induce neurobiological changes that further augment SUD risk [5].

There are many factors associated with adolescent drug and alcohol experimentation and abuse, such as personal values [6] and personality traits [7,8]. Among these factors, one of the most important is parental style. Maccoby and Martin developed a theoretical model of parenting styles with two fundamental dimensions in parenting practices: demandingness (strictness, imposition, parental firmness) and responsiveness (warmth, acceptance, involvement) [9]. Four parenting styles are defined according to these dimensions: authoritative, neglectful, indulgent and authoritarian. The term authoritative is used for the parenting style that combines high levels of control and affection. Parents with low responsiveness and few demonstrations of affection are classified as neglectful. Indulgent parents are affectionate, but demand little. Authoritarian parents are very demanding but demonstrate little affection, showing low levels of support for their adolescent children [10,11]. Studies have shown that adolescents are more likely to use drugs when they feel neglected by their parents [12,13]. At the same time, adolescents are more likely not to use drugs when they feel their parents have authoritative styles [10,11,14,15]. In particular, studies with Brazilian adolescents showed that not feeling supervised, and never feeling understood by parents were associated with illicit drug use [16]. Also, monitoring of the parents was an important predictor for the prevention of polydrug use among adolescents [17]. Thus, it has been proposed, that the authoritative parental style can be considered more protective and beneficial for the normal development of youngsters, resulting in good relationships, academic success, and positive psychosocial adjustment, [18] and is therefore likely to prevent drug-related problems. Recent data about indulgent parenting show that higher levels of self-esteem and satisfactory school performance of adolescents are also associated with this style [19,20,21]. Regarding the authoritarian style, a study showed that this are associated with low levels of self-esteem and high levels substance use [21].

On the other hand, the relationship between parental styles and the process of behavioral change toward stopping the use of different types of substances by adolescents remains insufficiently studied. Therefore, information about the style of the parents or of the individuals with a parental role and its influence on behavioral change, mainly the cessation of substance use by their children, is important for the development of treatment strategies for special populations. As the authoritative style can provide a higher level of connection between children and parents and is associated with a protective role against initial drug use [10], our hypothesis was that Brazilian adolescents’ perception of an authoritative parenting style is associated with substance use cessation. To address this hypothesis, this study aimed to verify the association between parental styles and behavioral changes among 14- to 19-year-old adolescents regarding their consumption of alcohol, tobacco, cannabis, cocaine/crack over one month of follow-up.

## 2. Materials and Methods

### 2.1. Subjects

The sample consisted of 99 adolescents between 14 and 19 years of age, of whom 60.6% were boys, 44.3% were students, 49.5% had nine or more years of education, and 60.4% were in a household with an income lower than 1 and a half minimum wage (Brazilian real). The participants were selected by meeting the following inclusion criteria: (a) all adolescents who called and completed the demandingness and responsiveness parental scale [22] during the 30 days of follow-up; (b) those who provided informed consent to take part in the research through verbal acceptance after the research objectives and purposes were explained.

Verbal acceptance was duly recorded in the computer protocols and tacit acceptance of the protocols was assumed as the youngsters maintained telephone contact with the researchers [23]. Adolescents were excluded if they (a) did not accept taking part in the study, (b) reported being under the influence of some substance during the call, (c) did not complete the 30-day follow-up, and (d) did not fill out the parental demandingness and responsiveness [22].

### 2.2. Procedure

A follow-up was carried out with a sample of adolescents who reported consuming alcohol and other drugs and who dialed in to the call center—the National Service for Guidance and Information on Drug Use (Ligue 132). This call center provides counseling and information on psychoactive substances to the entire Brazilian population free of charge while ensuring caller anonymity [24,25]. The participants in this research received a letter and educational material with information about the drug of abuse and behavioral change [26]. The letter had the dates of return calls to the service in 1, 3, 7 and 30 days after the cessation attempt. Here, we used only the 30-day follow-up data.

The data were collected between March 2009 and October 2015 by undergraduate students who had been trained for a total of 150 h in the use of a neuroscience-based educational model for the prevention and intervention in drug abuse [24]. The students also received specific training in brief motivational intervention, psychoactive substance consumption, parent/child relationships, and parenting styles.

### 2.3. Questionnaire

A sociodemographic characteristic questionnaire [3] was completed and included information about sex, age, family income, occupation, and help seeking prior to contacting the service. For the assessment of disorders due to psychoactive substances, we used a questionnaire created by the National Household Survey on Drug Abuse (NHSDA) [27], which classifies the presence of disorders as mild, moderate, and severe, according to criteria by the Statistical Manual of Mental Disorders DSM-5 [28]. According to this instrument, the diagnosis of drug abuse is made when the user meets at least two of the following criteria: (a) spends a lot of time getting drugs, using them, or recovering from their effects; (b) uses drugs in greater amounts or with greater frequency than desired; (c) shows tolerance (needs to use a greater amount of the substance to produce the same effect); (d) was in a situation of physical risk under the effect or after the effect of drugs; (e) had personal problems due to substance use (with family, police or school, emotional or psychological difficulties); and (f) wishes to reduce or discontinue substance use. Substance consumption was assessed through questions that characterized the amount used per event, frequency, and consumption pattern. This questionnaire was based on one from the World Health Organization (OMS), which has been widely employed in Brazil [3].

The Contemplation Ladder [29,30] was used to assess motivation for behavior change. One example of the questions used is “I’m doing something to stop using alcohol” (action stage). The stages are based on five statements that are an indication of the motivational stages of the adolescents, i.e., precontemplation, contemplation, preparation, action, and maintenance toward changing consumption behavior. The Contemplation Ladder has been adapted to and validated in Brazilian Portuguese [30]. The adolescents were considered to have changed behavior when they reported having ceased or decreased substance use at the 30-day follow-up with Ligue 132. Consumption decrease was analyzed along with cessation in view of some studies that report the initial goal of providing care to adolescents is to reduce drug consumption, while abstinence is approached later [31]. 

The parental demandingness and responsiveness scale was used to assess parenting styles. The instrument consists of two scales that are proposed to measure the orthogonal dimensions of demandingness and responsiveness [22]. In this specific study, the dimensions are not orthogonal, although theoretically they are, as is the case for other well know measures of parenting (as reviewed by Martinez [9]). Pearson correlation was applied, showing that there was correlation between demandingness and responsiveness; this was a moderate correlation among the mothers (*r* = 0.46; *p* < 0.001) and a strong correlation for the fathers (*r* = 0.69; *p* < 0.001). The outcome was used to define four parenting styles (authoritarian, authoritative, neglectful, and indulgent) based on Maccoby and Martin’s theoretical model [18]. This instrument comprises a five-point Likert scale with items ranging from never to always, and allows the assessment of parental styles based on the perception of the adolescents. It contains 24 items, 12 of which relate to demandingness and 12 to responsiveness, assessed from the standpoint of the adolescent and based on the educational practices of the father and mother, who were separately assessed. The items associated with demandingness refer to parenting practices of control and supervision of children’s activities, expressed in items such as: “Controls your school grades”. The responsiveness items denote the transmission of affection, support and involvement in the relationship between parents and children, as in the following item: “You can count on your father’s/mother’s help in case you have some problem”. The four types of parental styles were obtained based on the calculation of medians of the sample for the subscales of parental demandingness and responsiveness. The median values obtained in this study were 32 for demandingness and 35 for maternal responsiveness, 26 for demandingness and 27 for paternal responsiveness. The parenting styles were classified into four categories, based on the combination of these two dimensions, considering the median values cited as the cutoff point: authoritative (parents scoring above median in demandingness and responsiveness), authoritarian (those scoring above median in demandingness and below median in responsiveness), indulgent (parents scoring below median in demandingness and above median in responsiveness), and neglectful (those scoring below median in both demandingness and responsiveness). The authors of the scale suggest excluding subjects with the same median values in the parental demandingness and responsiveness subscales, which was done in this study [22]. Seven participants were excluded for this reason, from a sample of 106 adolescents who answered the scale. Cronbach’s Alpha was calculated to evaluate the internal consistency of the instrument considering their subscales. The instrument obtained a Cronbach Alpha for demandingness of 0.79 for the maternal figure and 0.75 for the paternal figure, and Cronbach Alpha for responsiveness of 0.92 and 0.89, respectively. The presence of a maternal or paternal figure was surveyed using the parental demandingness and responsiveness scale. When the adolescent reported not having contact with the father, mother, or someone who he considered as a substitute of the paternal or maternal figure, it was considered that this figure was not present in their lives. In the absence of a father and/or mother, the parental style of foster parents (grandparents, uncles and aunts, or other relatives) was assessed. Some adolescents did not answer the items for paternal or maternal style as they did not have a person fulfilling such a role in their lives.

Parental substance use history was observed by asking the adolescents: “Does anyone in your family have problems related to the use of alcohol, tobacco, and/or other drugs?” In case of an affirmative answer, the adolescent was asked which family member used.

### 2.4. Design

This is a prospective study with young people extracted from of a randomized controlled trial (RCT). The sample an excerpt of a RCT was randomized to receive one of two types of intervention: (1) a brief motivational intervention with a duration of approximately 45 min that sought to promote motivation and self-efficacy and identify risk factors and coping strategies, and planning to change substance use behavior; (2) a minimal intervention focused on providing information on the substances and providing support though the educational material sent and meeting the ethical standards according to which all subjects be treated with some type of intervention. The comparison of the two interventions did not show differences in the variables assessed in the current study.

### 2.5. Data Analysis

Descriptive statistical analyses were performed for the demographic variables, consumption characteristics, stages of change, substance abuse by the parents, and parental styles perceived by the adolescents. A chi-squared test was used to verify associations between categorical variables and behavior change. The dependent variable considered was the changed behavior of the substance consumption at 30 days of follow-up (alcohol, tobacco, cannabis or cocaine/crack). Response categories were dichotomized: (no/yes). To reduce confounding factors between behavior change toward substance use and the variables studied, a Multivariate analysis by Poisson regression was carried out, with a robust variance estimator to calculate the Relative Risk (RR) and a 95% confidence interval (95% CI). The variables with *p* < 0.20 in the bivariate analysis were selected and included in the multivariate analysis. Considering these criteria, the occurrence or not of change in the consumption of each substance at 30 days of follow-up. The independent variables for changed alcohol consumption were maternal parental style (indulgent/authoritative) and motivation (precontemplation, contemplation and preparation/action and maintenance), while paternal presence in the life of the adolescent (no/yes) was the independent variable for changed cocaine/crack consumption (no/yes). These variables were input into the regression model using the forward stepwise selection procedure. The analyses were performed in the software IBM SPSS version 19.0 (IBM Corp., Armonk, NY, USA) and *p* values below 0.05 were considered significant.

### 2.6. Ethics Procedures

The study was approved by the Research Ethics Committee of the Universidade Federal de Ciências da Saúde de Porto Alegre (UFCSPA) under process 451/09 and protocol 805/09.

## 3. Results

The demographic characteristics of the participants are shown in Table 1. Eighty-three percent of the sample reported substance use. The results indicated that 83.7% of the adolescents had consumed alcohol and 74.7% had consumed tobacco. The consumption of cannabis and cocaine/crack corresponded to 65.7% and 51.4%, respectively. Most of the adolescents had been consuming the analyzed substances for over one year. Tobacco and cannabis were more often consumed daily, whereas alcohol and cocaine/crack were consumed weekly or less often. Most of the subjects in the sample had moderate disorders regarding substance use (Table 2).

The majority of these young individuals had previously sought help prior to contacting the “Ligue 132” program (76.8%). Parental substance use behavior showed that the most mothers of the adolescents did not use substances (70.2%), unlike the fathers, of whom 58.1% used some kind of drug (Table 1). Despite the higher rate of drug use by the fathers, this parental behavior did not statistically influence the outcomes of the adolescents. Out of a total of 99 adolescent participants, 90 responded with a perceived the maternal style and 77 with a paternal style. The most prevalent parental styles reported by the adolescents were authoritative maternal style (28.9%) and neglectful paternal style (42.9%) (Table 3).

The adjusted analysis indicated that the adolescents who perceived an indulgent style in their mothers were more likely not to change their behavior toward alcohol use within the 30-day follow-up (RR 5.1; 95% CI 1.7–14.8). The adolescents whose fathers (or paternal figures) were not present in the family routine were more likely not to change their behavior related to cocaine/crack consumption within the 30 days of the follow-up (RR 3.4; 95% CI 1.2–9.8) (Table 3). Therefore, parental styles did not have an important overall effect on stopping or decreasing licit or illicit drug abuse by adolescents.

## 4. Discussion

The relationship between behavior change and parental styles is poorly investigated. The most important contribution of this study is the investigation of the association between parental styles and changes in problematic substance abuse behavior by adolescents who already use drugs. In contrast, previous studies primarily assessed the influence of parental styles on the initiation of drug abuse [10,11,14,15,17]. Overall, the literature has shown that when both parents monitor adolescents, they are less likely to use drugs than those reared with less parental supervision [14,15,32]. More specifically, youngsters who do not use drugs more often report authoritative behaviors of their mothers [10,14]. In the present study, with adolescents that who are already presenting drug abuse problems, the parenting style seems to be of modest influence for adolescent’s drug use behavioral changes. For the alcohol consumption, our results show that authoritative mothers seem to be a positive factor for stopping or reducing alcohol use by their children, while indulgent mothers seem to act as hurdles for decreasing drug abuse behaviors of their progeny. Indulgent parents have been acknowledged as being more tolerant, with the highest expression of affection, while exerting little authority, and making few demands for the mature behavior of their children [18,33], and therefore, there may be little demand for behavior change [15]. A study that investigated the progression of alcohol use among youngsters showed an increase in alcohol consumption over time among adolescents whose mothers were more tolerant in their communication regarding alcohol [34]. Moreover, favorable attitudes by the parents toward adolescent’s alcohol consumption and parental drinking were negatively associated with parental monitoring, quality of relationships between parents and child, support and parental involvement [35]. When there is little demand in the interaction with their parents, the adolescents perceive greater permission toward any behavior, including substance abuse and other risk behaviors [36]. It calls ones attention that indulgent behavior might not always be permissive towards defiant or risky behaviors, since greatest personal well-being was found for adolescents raised with higher indulgent parenting style and the greatest social well-being was found for adolescents raised with higher parental warmth, by both indulgent and authoritative parents [37]. On the other hand, the authoritative parenting style is a protective factor, and the neglectful style is a risk factor for Brazilian youngsters in becoming polydrug users [17]. However, drug abuse by adolescents who are already drug consumers is influenced little by the previous indulgent or authoritative parental styles, especially when one considers the use of illicit drugs. The reasons for this lack of influence needs to be better detailed in the future.

Bares et al. showed that neglectful or absent paternal figures significantly contributed to substance use by their children [32]. Concomitantly, more participative fathers seemed to be a protective factor against the use of alcohol and drugs [38]. In the present study, not having a present father figure in life may be an important factor for maintaining the ongoing adolescents cocaine/crack abuse. However, the father’s presence was not significant in the behavioral changes regarding alcohol, tobacco, or marijuana. Several factors may interact to decrease the different risky or deviant behaviors in teenagers. When fathers are more involved in the life of their children, the adolescents will have higher self-esteem and a better perception of the fathers’ abilities. A healthy father–child relationship lowered psychological hardships and the risk of more severe involvement with psychoactive substances [11,38]. Additionally, herein we show that an absent paternal figure may be associated with lower chances of changing the behaviors of illicit drug (cocaine/crack) use.

On the other hand, it is important to keep in mind that the association of the factors concerning parental characteristics with change in behavior are not disconnected from a broader social context. Youngsters who consume substances are exposed to a range of physiological harm, are more vulnerable to committing criminal offenses, to living on the streets, to losing family ties and to having problems at school, as well as getting involved with drug trafficking [11,39]. This study did not find a significant association between the sociodemographic data and family history of substance use with change in behavior. However, an important issue to consider in our paper is the fact that most households (60.4%) are classified as low income. Interestingly, as low socioeconomic status families are more likely to live in hazardous communities where crime is higher, an authoritarian parenting style, for example, may not be as harmful in this environment, and it may even have some protective benefits [40]. On the other hand, horizontal collectivist cultures, such as Brazil, underscore egalitarian relations, and the use of affection, acceptance, and involvement in raising children is of greater focus [37]. A study conducted with a large sample of European adolescents found that regardless of the country, an authoritative parenting style and an indulgent parenting style (support without strictness and imposition to set limits) were equally protective against drug-use. However, the indulgent parenting style performed even better than the authoritative parenting style when examining the outcomes of self-esteem and school performance [19]. In this sense, recent results have shown that the presence of affection seems to be essential today in cultures such as Brazil. Additionally, as our sample has a low socioeconomic status and for the drug use context, some degree of parental strictness is fundamental. In this sense, for our sample and our focus, the authoritative style shows better results. Thus, we emphasize that the social context must always be analyzed in conjunction with the analysis of parenting styles.

The data showed a high prevalence of substance use by the fathers of the adolescents. Although there is no statistically significant correlation between parent drug use and persistent adolescent drug abuse, our results seem to agree with the literature. As already described, the family context serves as a model for children behaviors. Hence, parents who use psychoactive substances may negatively influence youngsters to do the same [41]. It is known that a dysfunctional environment models the type of behavior in which children might repeat the actions of their parents and may fail to change behavior regarding drug consumption, even when there are problematic drug abuse [42,43]. One may consider that the family’s context with indulgent behaviors may include family habits related to pro-substance consumption that negatively interfere for the behavior changes of the adolescents.

A limitation of this study was that the sample studied was small, because of the low retention of subjects in the protocol, which has proven to be a challenge in studies on SUD among adolescents. The challenges of treating adolescents might influence the impact of treatment in the outcome and more studies are important to be planned. Another limitation is related to the assessment of behavioral change 30 days after the intervention, considering that this seems to be a short period of the process, albeit a crucial one for the subsequent stages of change [44]. The multiple confounding factors regarding adolescents’ behaviors also need further investigation.

In short, this study shows an association between some of the mothers’ parenting characteristics, influence the adolescents to stop or decrease alcohol abuse, and having a competent and present father figure is important regarding the decrease of cocaine/crack abuse. Alcohol seems to be a substance with greater acceptance by families as it is licit and is usually consumed in the presence of the Brazilian parents [34,43], it is more readily available for purchase and is commonly consumed among friends on social occasions, which facilitates the acceptance of its use by the society [43]. However, mothers are important to help their siblings stop the deviant behaviors regarding alcohol abuse. Cocaine/crack, in contrast, is an illegal substance with intensively reinforcing pharmacological effects, which accounts for difficulties in achieving abstinence, facilitates the process of addiction, and hampers behavioral changes by the adolescents [45], and understandably requires the presence of a strong father figure. 

## 5. Conclusions

This study demonstrates that even though healthier parent–child relationships seem to have some influence on adolescents in stopping an individual’s drug abuse, the authoritative parental style is more important for overall drug abuse prevention. It is important to acknowledge that the adolescents’ perception of a more indulgent maternal style decreases the chances to change alcohol consumption in comparison to a more authoritative maternal style. The absence of the father in the family routine is an important factor for no change in cocaine/crack consumption. Although these associations clearly show that paternal presence and maternal authoritative styles influence whether adolescents consider and succeed in stopping substance use, other evaluations are still needed with longer follow-up.

## Figures and Tables

**Table 1 ijerph-16-03432-t001:** Sociodemographic characteristics of the adolescents (*n* = 99).

Characteristics	*n* (%)
**Gender** (*n* = 99)	
Girls	39 (39.4)
Boys	60 (60.6)
**Schooling** (*n* = 98)	
≤9 years of education	49 (49.5)
>9 years of education	49 (49.5)
**Occupation** (*n* = 97)	
Employed	31 (32)
Unemployed	23 (23.7)
Student	43 (44.3)
**Household income** (*n* = 91)	
≤1 and a half minimum wage	55 (60.4)
>1 and a half minimum wage	36 (39.6)
**Age** (*n* = 99) (mean ± SD)	17.05 ± 1.51
**Previous help to cease substance use** (*n* = 96)	
Yes	76 (76.8)
No	20 (20.2)
**Maternal substance use** (*n* = 84)	
Yes	25 (29.8)
No	59 (70.2)
**Paternal substance use** (*n* = 86)	
Yes	50 (58.1)
No	36 (41.9)
**Both parents substance use** (*n* = 84)	
Yes	34 (40.5)
No	50 (59.5)

Household income: value in Brazilian real (BRL); exchange rate in August 2019: USD 1 = BRL 3.97.

**Table 2 ijerph-16-03432-t002:** Substance consumption characteristics among the adolescents.

Consumption Characteristics	Substance							
	Alcohol		Tobacco		Cannabis		Cocaine/Crack	
	*n*	%	*N*	%	*n*	%	*N*	%
**Consumption in the last year**								
Yes	77	77.8	72	72.7	59	59.6	67	67.7
No	22	22.2	27	27.3	40	40.4	32	32.3
**Consumption in the last month**								
Yes	72	72.7	68	68.7	53	53.5	54	54.5
No	27	27.3	31	31.3	46	46.5	45	45.5
**Years of use**								
Up to a year	18	23.4	21	29.2	21	35.6	24	35.8
Over a year	59	76.6	51	70.8	34	64.4	43	64.2
**Frequency of use**								
Daily	13	18.3	61	89.7	28	52.8	34	43.6
Weekly/less than weekly	58	81.7	7	10.3	25	47.2	44	56.4
**Usual amount consumed/day**								
Up to 5 (doses/cigarettes/grams/rocks)	15	20.8	18	26.5	46	86.8	34	45.9
Over 5 (doses/cigarettes/grams/rocks)	57	79.2	50	73.5	7	13.2	20	54.1
**Substance use disorder**								
No disorder	11	14.3	15	20.8	8	13.6	11	16.4
Mild	19	24.8	18	25.0	17	28.8	10	14.9
Moderate	36	46.6	30	41.7	24	40.7	25	37.3
Severe	11	14.3	9	12.5	10	16.9	21	31.4

Data are presented as n and percentage.

**Table 3 ijerph-16-03432-t003:** *Relative Risk* (RR) for substance use behavior change in the 30-day follow-up.

Variables	30-Day Follow-up													
	Alcohol				Tobacco			Cannabis			Cocaine/Crack			
	Changed Behavior				Changed Behavior			Changed Behavior			Changed Behavior			
	No	Yes	Raw	Adjusted	No	Yes	Raw	No	Yes	Raw	No	Yes	Raw	Adjusted
	*n* (%)	*n* (%)	RR (95% CI)	RR (95% CI)	*n* (%)	*n* (%)	RR (95% CI)	*n* (%)	*n* (%)	RR (95% CI)	*n* (%)	*n* (%)	RR (95% CI)	RR (95% CI)
**Treatment**														
Minimum Intervention	10 (22)	35 (78)	1.1 (0.4–3.7)		13 (31)	29 (69)	1.0 (0.4–3.0)	7 (21)	26 (79)	1.1 (0.3–4.5)	8 (24)	26 (76)	1.7 (0.4–7.5)	
IBM	6 (10)	25 (81)			8 (30)	19 (70)		4 (19)	17 (81)		3 (15)	17 (85)		
**Motivation**														
PC/C/P	6 (29)	15 (71)	2.1 (0.6–7.5)	1.8 (0.7–4.7)	10 (40)	15 (60)	2.0 (0.7–6.0)	4 (40)	6 (60)	3.3 (0.7–14.9) ^+^				
A/M	7 (16)	38 (84)			9 (25	27 (75)		7 (17)	35 (83)					
**Maternal style**														
Neglectful	2 (12)	15 (88)	1.2 (0.1–9.6)		4 (25)	12 (75)	0.9 (0.2–4.3)	4 (29)	10 (71)	1.6 (0.3–8.9)	2 (18)	9 (82)	0.9 (0.2–5.4)	
Authoritative	2 (10)	18 (90)			5 (26)	14 (74)		3 (20)	12 (80)		7 (19)	30 (81)		
**Maternal style**														
Indulgent	8 (40)	12 (60)	**6.0 (1.1–33.3) ***	**5.1 (1.7–14.8) ***	7 (44)	9 (56)	2.2 (0.5–9.0)	1 (9)	10 (91)	0.4 (0.04–4.5)	2 (12)	14 (88)	0.5 (0.09–2.8)	
Authoritative	2 (10)	18 (90)			5 (26)	14 (74)		3 (20)	12 (80)		7 (22)	25 (78)		
**Maternal style**														
Authoritarian	3 (21)	11 (79)	2.4 (0.3–17.1)		3 (25)	9 (75)	0.9 (0.2–4.9)	1 (11)	8 (89)	0.5 (0.04–5.7)	3 (37)	5 (63)	3.4 (0.6–18.1)	
Authoritative	2 (10)	18 (90)			5 (26)	14 (74)		3 (20)	12 (80)		6 (15)	34 (85)		
**Paternal style**														
Neglectful	4 (16)	21 (84)	0.5 (0.1–2.5)		5 (22)	18 (78)	0.5 (0.1–2.3)	4 (27)	11 (73)	0.9 (0.2–5.6)	1 (6)	15 (94)	0.3 (0.03–2.6)	
Authoritative	4 (27)	11 (73)			4 (36)	7 (64)		3 (27)	8 (73)		5 (19)	21 (81)		
**Paternal style**														
Indulgent	2 (14)	12 (86)	0.4 (0.07–3.0)		8 (53)	7 (47)	2.0 (0.4–9.8)	0 (0)	12 (100)	1.4 (0.9–1.9)	1 (11)	8 (81)	0.7 (0.07–7.1)	
Authoritative	4 (27)	11 (73)			4 (36)	7 (64)		3 (27)	8 (73)		5 (14)	29 (86)		
**Paternal style**														
Authoritarian	1 (20)	4 (80)	0.6 (0.05–8.1)		0 (0)	3 (100)	1.5 (1.0–2.4)	0 (0)	3 (100)	1.3 (0.9–1.9)	1 (50)	1 (50)	7.2 (0.3–134.2)	
Authoritative	4 (27)	11 (73)			4 (36)	7 (64)		3 (27)	8 (73)		5 (12)	37 (88)		
**Maternal presence**														
No	1 (20)	4 (80)	0.9 (0.09–8.9)		2 (33)	4 (67)	1.1 (0.1–6.8)	2 (40)	3 (60)	2.9 (0.4–20.4)	2 (33)	4 (67)	2.2 (0.3–13.7)	
Yes	15 (21)	56 (79)			19 (30)	44 (70)		9 (18)	40 (82)		9 (19)	39 (81)		
**Paternal presence**														
No	5 (29)	2 (71)	1.9 (0.5–6.2)		4 (24)	13 (76)	0.6 (0.2–2.2)	4 (31)	9 (69)	2.1 (0.5–9.0)	5 (45)	6 (55)	**5.1 (1.2–22.3) ***	**3.4 (1.2–9.8) ***
Yes	11 (19)	48 (81)			17 (33)	35 (67)		7 (17)	34 (83)		6 (14)	37 (82)		

Some subjects did not answer all the questions: 95% CI = 95% confidence interval; RR = relative risk; PC = precontemplation; C = contemplation; P = preparation; A = action; M = maintenance; * *p* < 0.05. ^+^ Motivation for cannabis, and cocaine/crack were assessed jointly using the motivation ladder for illicit substances.

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
