# Peer review of "Role of Parenting Styles in Adolescent Substance Use Cessation: Results from a Brazilian Prospective Study"

_ijerph, 2019, doi:10.3390/ijerph16183432_

Round 1

Reviewer 1 Report

The study analyzes the relation of parenting styles with adolescents’ changes in substance consumption. Although the subject of the study is interesting the work present important deficiencies that should be corrected

ABSTRACT

Information about the sample size, the age and the gender of the participants must be given but must not been situated in the results.

Data from the analysis should not be included in the abstract.

INTRODUCTION

The introduction of the paper must be applied. The revision about parenting is very limitated. More recent studies about parenting influence in adolescent must been include in the theoretical review (e.g. Calafat, García, Juan, Becoña, & Fernández-Hermida, 2014). Specially those carry out in Brazil (e.g. Martínez & García, 2008), that have shown that the benefits of indulgent parenting are similar o even higher to authoritative parenting, along with a increasing number of studies recently published (e.g. see the article published in this IS : Researching parental socialization styles across three cultural contexts: Scale ESPA29 bi-dimensional validity in Spain, Portugal, and Brazil (Martínez, García, Fuentes, et al.). T

On the other hand, other factors associated with substance consumption must been at least mentioned, such as personal values (Galdós & Sánchez, 2010),  personality traits (Saiz, Álvaro & Martínez, 2011; Liraud, & Verdoux, 2000).

Some hypothesis of the expected direction of the relation between parenting and substance changes behavior must be formulate and justified.

METHOD

The order of the method section is confusing. It must be as follows: Subject/ Procedures/ Questionnaire/ Design

In the Design section must not been included information of the Procedure section.

In the Subjects section information about the age, sex and demographic characteristic of the participants, such as racial origin, and others socio-economic characteristic, must been included.

Information about the number of participants that inform from the mother, father or both parents’ behavior also must be provided.

The implications and the reasons of the exclusion of parents with same average scores in the parental demandingness and responsiveness subscales must been explained. One can deduce that are the indulgent parents who are excluded. This question must be clarified.

Information about of the questionnaires used in the study must be extended. Items examples of all the scales must been provided.

The descriptive analysis of the demographic variables must not be treated as a result. Most of this information must appear in the description of the sample section, since it is not a result but information of the study sample.

The description of the data analysis is confusing. The authors must explain better why they chose regression instead to MANOVA in other to compare the results of each parental style on the dependent variables, since the sample have classified in the four parenting styles.

DISCUSSION

The inconsistency of this results with previous studies in Brazil must been explained. It seems very important to me to consider the high prevalence of drug use by adolescents’ fathers, and this would inform of this result. This can situate this family’s in a context where indulgent style or authorities parenting can fail because the family norm is “substance consumption”.

MINOR QUESTIONS

Correct reference on section 2, line 48.

The first sentence section 4 is incorrect. The words “the influence” must been deleted.

REFERENCES:

Calafat, A., García, F., Juan, M., Becoña, E., & Fernández-Hermida, J. R. (2014). Which parenting style is more protective against adolescent substance use? Evidence within the European context. Drug and alcohol dependence138, 185-192.

Martinez, I., & Garcia, J. F. (2008). Internalization of values and self-esteem among Brazilian teenagers from authoritative, indulgent, authoritarian, and neglectful homes. ADOLESCENCE-SAN DIEGO-, 43(169), 13.

Martínez, I., Garcia, F., Fuentes, M. C., Veiga, F., Garcia, O. F., Rodrigues, Y., ... & Serra, E. (2019). Researching Parental Socialization Styles across Three Cultural Contexts: Scale ESPA29 Bi-Dimensional Validity in Spain, Portugal, and Brazil. International journal of environmental research and public health, 16(2), 197.

Galdós, J. S., & Sánchez, I. M. (2010). Relationship between cocaine dependence treatment and personal values of openness to change and conservation. Adicciones, 22(1), 51-58.

Saiz, J., Álvaro, J. L., & Martínez, I. (2011). Relation between personality traits and personal values in cocaine-dependent patients. Adicciones, 23(2), 125-132.

Liraud, F., & Verdoux, H. (2000). Which temperamental characteristics are associated with substance use in subjects with psychotic and mood disorders?. Psychiatry Research93(1), 63-72.

Author Response

Dear Reviewer 1,

We are submitting a revised version of our manuscript entitled “Role of parenting styles in adolescent substance use cessation: results from a Brazilian longitudinal cohort study”. We have modified the paper in response to the extensive and insightful reviewers comments. In this revised -version, we have addressed the concerns of the editor and the reviewers. Finally, we would like to thank the reviewers for their valuable collaboration in improving the quality of our manuscript.  We listed all changes according to the reviewer 1 suggestions below (marked red color in the revised manuscript):

Response to Reviewer 1 Comments

ABSTRACT

Information about the sample size, the age and the gender of the participants must be given but must not been situated in the results.

Data from the analysis should not be included in the abstract.

Thank you for your comment. We did these changes in the abstract.

INTRODUCTION

The introduction of the paper must be applied. The revision about parenting is very limitated. More recent studies about parenting influence in adolescent must been include in the theoretical review (e.g. Calafat, García, Juan, Becoña, & Fernández-Hermida, 2014). Specially those carry out in Brazil (e.g. Martínez & García, 2008), that have shown that the benefits of indulgent parenting are similar o even higher to authoritative parenting, along with a increasing number of studies recently published (e.g. see the article published in this IS : Researching parental socialization styles across three cultural contexts: Scale ESPA29 bi-dimensional validity in Spain, Portugal, and Brazil (Martínez, García, Fuentes, et al.).

We are in agreement. We included these references in the end of the second paragraph of the introduction.

On the other hand, other factors associated with substance consumption must been at least mentioned, such as personal values (Galdós & Sánchez, 2010),  personality traits (Saiz, Álvaro & Martínez, 2011; Liraud, & Verdoux, 2000).

Thanks for pointing out. We included these aspects in the beginning of the second paragraph.

Some hypothesis of the expected direction of the relation between parenting and substance changes behavior must be formulate and justified.

We are in agreement. We included an hypothesis in the end of the last paragraph of the introduction.

METHOD

The order of the method section is confusing. It must be as follows: Subject/ Procedures/ Questionnaire/ Design

The order was altered.

In the Design section must not been included information of the Procedure section.

Thanks for pointing out. We changed the information in the both sections.

In the Subjects section information about the age, sex and demographic characteristic of the participants, such as racial origin, and others socio-economic characteristic, must been included.

Well, about this, we did not collect the racial information of the participants. We added information in the Subjects section about the socio-characteristics that were collected (schooling, household income, occupation, and previous help).

Information about the number of participants that inform from the mother, father or both parents’ behavior also must be provided.

We included this information in the Table 1.

The implications and the reasons of the exclusion of parents with same average scores in the parental demandingness and responsiveness subscales must been explained. One can deduce that are the indulgent parents who are excluded. This question must be clarified.

The reason of the exclusions of the subjects occurs when the score of this participant is equal of the median obtained in the parental demandingness and responsiveness subscales. In this case, we did not to define which exact parental style.  The implication is that we will ensure that any result will be inferred.

Also we included the values of the median of these subscales in the fourth paragraph of the Questionnaire of the Materials and Methods section.  

We put in color green the phrase that explained the suggestion of the authors of the scale (second paragraph of the Subjects section).

Information about of the questionnaires used in the study must be extended. Items examples of all the scales must been provided.

We are in agreement. We included more information throughout the questionnaire session.

The descriptive analysis of the demographic variables must not be treated as a result. Most of this information must appear in the description of the sample section, since it is not a result but information of the study sample.

The only sociodemographic information that was an inclusion criteria was the age of the participants. However, the descriptive analysis of the demographic variables was included in the first paragraph Subjects section.

The description of the data analysis is confusing. The authors must explain better why they chose regression instead to MANOVA in other to compare the results of each parental style on the dependent variables, since the sample have classified in the four parenting styles.

The variables dependent on our study are not continuous. Therefore, we decided to use an analysis for categorical and dichotomous variables, such as logistic regression analysis. To improve data analysis, we redid the process. We also chose to do the Poisson regression analysis, a generalized linear model, more suitable for data analysis.

DISCUSSION

The inconsistency of this results with previous studies in Brazil must been explained. It seems very important to me to consider the high prevalence of drug use by adolescents’ fathers, and this would inform of this result. This can situate this family’s in a context where indulgent style or authorities parenting can fail because the family norm is “substance consumption”.

Thank you for your suggestion. We included more Brazilian studies data in the Discussion section.

MINOR QUESTIONS

Correct reference on section 2, line 48.

Thanks for pointing out. The change was made.

The first sentence section 4 is incorrect. The words “the influence” must been deleted.

We agree. The error was solved.

Reviewer 2 Report

Comments for authors attached as a word document.

Author Response

Dear Reviewer 2,

We are submitting a revised version of our manuscript entitled “Role of parenting styles in adolescent substance use cessation: results from a Brazilian longitudinal cohort study”. We have modified the paper in response to the extensive and insightful reviewers comments. In this revised -version, we have addressed the concerns of the editor and the reviewers. Finally, we would like to thank the reviewers for their valuable collaboration in improving the quality of our manuscript.  We listed all changes according to the reviewer 2 suggestions below (marked blue color in the revised manuscript):

Response to Reviewer 2 Comments

I am not entirely sure that authors have provided results from a longitudinal study. The analysis follows a cross-sectional study design, with the outcome coded at the end of the 30-day follow-up. Other variables included are not constructed as change over time variable. Authors should reconsider the title and description of their study.

Thank you for your suggestion. We have reviewed these aspects and indeed the study design is best characterized as a prospective study, with follow-up of 30-days. We change that did report in the Design Sections.

I am not entirely convinced if tobacco use should be compared with alcohol and illicit drug use. Authors should explain their reasoning for including tobacco consumption in the beginning of the paper. There is also a lot of variability in how adolescents consume both licit and illicit substances. The paper would benefit from authors focusing on either alcohol use or illicit substance use.

The choice to include tobacco consumption has been to cultural issues. In Brazil, tobacco use is viewed as harmful as alcohol use. In addition,  tobacco consumption as well as alcohol use reduces self-control and increases risk behaviors for this reason sounds interesting considered this users (WHO, 2016).

World Health Organization (WHO) . Media Centre: Health for the World’s Adolescents a second chance in the second decade. Geneva: WHO; 2014. http://www.who.int/mediacentre/news/releases/2014/focus-adolescent-health/en/

The authors note that the study includes a randomized clinical trial component in terms of the treatment provided to adolescents who followed-up after 30 days, but these treatments do not have a significant association with behavior change. The implication of this result is not discussed at all in the discussion or conclusion section. What is the independent impact of treatment on outcome (i.e., behavior change), and is it associated with the parental styles of the adolescents?

As that was not the main focus of study no described in the results section but we included this as a limitation of study associated with small sample.

Is there any way to control for adolescents who consume multiple substances at the same time? Would that impact the results of the study?

Substantive clarifications— 1. In lines 23 and 24 of Section 1(Introduction), authors mention that “parents and other family members” have an impact on behavioral change in adolescent substance use, but it seems that other family members here include any individual with a parental influence (as noted in section 2.3). Authors should clarify this in the beginning.

Thanks for pointing out; we agree. We change the phrase that did report “other family members” in the last paragraph of the introduction.

Authors should clarify the differences in parenting styles as opposed to only mentioning it as they do now. Not all readers may be familiar with differences in parenting styles. Some parenting styles are discussed briefly in the discussion section, but all styles included and analyzed in the study should be clarified in the introduction.

Thanks for pointing out; we agree. We included the description of the parental styles used in this study in the second paragraph of the introduction (red color).

Authors use drug use and alcohol use interchangeably throughout, it might be better to stick with “substance use” since the study looks at both alcohol and drug use.

Thanks for pointing out. The changes were made throughout the text.

Authors should provide more clarity on the coding for the dependent variable. Authors write in Section 2.5 (Data analysis), that “the occurrence or lack of change toward the consumption of these drugs at 30 days of follow-up was used as the dependent variable”. What was the reference category for the binary DV? In the results, authors interpret likelihood of “no change” but this is not clarified in the methods section or in table 3. Relatedly, authors should consider re-writing their data analysis section to explain how they have created their variables for the analyses using the items in the questionnaire.

In this aspect we consider change of substance consumption as reference for the binary DV. Some adjusts has take in the Data Analysis section.

What is the total sample size for the logistic regression analyses in Table 3? Were any cases dropped?

We included this information in the second paragraph of the results section.

The small sample size is an issue in terms of statistical power of the results too. Perhaps, the authors can draw on one or two case examples based on any narratives made by the researchers who followed-up with the sample of adolescents? This would further bolster the authors’ results.

Unfortunately we have no this information because this is a proposal of a qualitative study. But this limitation is described in the discussion section.

Authors note that they did not find any significant associations for crack use—this may be directly related to the actual numbers, which has the lowest proportion in the sample. Authors may consider if it is meaningful to include crack use in the study.

We opted for united the crack and cocaine users since the crack is cocaine derived.

Authors conclude that “more indulgent maternal style perceived by the adolescents increased the difficulty in ceasing or decreasing alcohol use. The absence of the father in family routine is an important factor for maintaining behavior toward cocaine consumption over 30 days” – but this is not as clear due to the way the dependent variable is presented by the authors.

We accepted your suggestion and  change this excerpt in the conclusion section.

Some minor grammatical changes, including— - On line 23 in section 1(Introduction), authors begin with “In lieu of this”. It should read as “In light of this”.

According the reviewer 1 suggestions, we change this part of the introduction and this is clearer now.

Lines 4-8 read in section 2.3 (Questionnaire) read as bullet points as opposed to full sentences. Can be re-worded as “Questionnaire elicited information about…”

This excerpt was modified according to reviewer 1 suggestions. 

In section 3 (Results), Line 20, should read as “Eighty-three percent of the sample reported substance use…”

We agree. The error was solved.

In section 3 (Results), Line 28, it should say “adolescents’ mothers” or “mothers of adolescents” instead of “adolescent mothers”

We agree. The error was solved.

Reviewer 3 Report

Present paper aims to investigate the contribution of parenting styles (indulgent, authoritative, authoritarian, and neglectful) in adolescent adjustment captured by drug use. Sample was 99 adolescents (n females = 55; 39.4%), ranging from 14 to 19 year-old (M = 17.05, SD = 1.51). Overall, it seems that the most protective parenting style against drugs use could be authoritative.

Present paper have important theoretically and methodological weakness that should be addressed before to consider its potential for parenting literature debate. Present paper, in the actual state, is quite difficult to understand what authors conducted and which results obtained in order to future replications of this study. The writing was convoluted and it was challenging to pick out the main findings of the study.

(1) Parenting styles and adolescent school adjustment.

(1a) Authors should review the theoretical assumptions of Maccoby and Martin’ theoretical framework (1983) about the orthogonality of parental dimensions (warmth and strictness). Also, how authors test whether their measures of parental are non-related or orthogonal? Authors should indicate this point examining the relationship between both parental measures.

(1b) Authors should review more detailed studies examining parenting styles and developmental outcomes across the globe. Importantly, authoritative parenting is commonly related to optimal adjustment among adolescents, according to findings from studies most of them conducted in middle-class white families from the United States (Darling & Steinberg, 1992; Steinberg, Elmen, & Mounts, 1989; Steinberg, Lamborn, Darling, Mounts, & Dornbusch, 1994). Nevertheless, studies examining parenting styles and adolescent compentence and adjusment in other cultural context indicate that the authoritative parenting styles is not always related to the optimal adjustment (Garcia & Gracia, 2009; Garcia, Lopez-Fernandez, & Serra, 2018; Darling & Steinberg, 1992; Pinquart, 2018). For example, as Darling and Steinberg noted (1992, p 487): “Particularly pressing issues are the variability in the effects of parenting style as a function of the child's cultural background”. And more currently, Pinquart (2018, p. 95) explained that “Associations of parenting styles with child outcomes may vary, to some extent, between cultural contexts, such as ethnic groups within countries and different regions of the globe”. On the one hand, some findings from studies with ethnic minority families and dangerous communities seem to suggest that authoritarian parenting may not be as harmful and may even have some protective benefits (Baumrind, 1972; Chao, 1994, 2001; Dwairy & Achoui, 2006; Dwairy, Achoui, Abouserfe, & Farah, 2006). On the other hand, some studies conducted with families from in European and Latin American countries seem to suggest that indulgent parenting is related to equal or even better developmental outcomes than those raised by authoritative parents (Garcia & Gracia, 2009; Garcia & Serra, 2019; Pinquart, 2018).

(1c) Authors should review studies in Brazil examining parenting styles and adolescent adjustment and competence. For example, some studies from Brazil (Martínez & García, 2008; Martínez, García, & Yubero, 2007) and Portugal (Rodrigues, Veiga, Fuentes, & García, 2013) seem to indicate that adolescents with indulgent parents obtained equal or even better scores than their peers with authoritative parents, whereas authoritarian and authoritative parenting styles are associated with the lowest scores.

(1d) Authors should add more studies examining parenting styles and drug use outcomes. For example, and interesting study examining parenting styles (indulgent, authoritative, authoritarian, and neglectful) and different indicators of competence an adjustment (including drugs) across five European-countries as Sweden, United Kingdom, Spain, Portugal, Slovenia, and the Czech Republic (Calafat, García, Juan, Becoña, & Fernández-Hermida, 2014) or parenting styles with a multidimensional criteria of drugs (Riquelme, García, & Serra, 2018).Or more currently, in Brazil, Juliana Valente and her colleagues (Valente, Cogo-Moreira, & Sanchez, 2017) have analyzed parenting styles (indulgent, authoritative, authoritarian and neglectful) in order to examine patterns of drug use in adolescence.

(2) Authors should indicate very detailed the family background of participants. Are middle-class families? How author control the key role of socioeconomic status? Information provided by authors (see Table 1) indicated that 60.4% adolescents of adolescents are from families with 1.500,00 Brazilian real (BRL); and those from families with more than 1.500,00 Brazilian real (BRL) represented 39.6%. Why authors have used 1.500,00 Brazilian real (BRL) as criteria? How authors take into account the variable socioeconomic status when they have classified adolescent into one of the four typologies?

(3) Measures section is difficult to understand. This point is very important in order to be possible replicate this study. Author should indicate clearly the variable and its specific measure. Authors should provide the items that they used for each measure in a Table. Importantly, also, authors should indicate alpha values for the subscales. For example, alpha values for demandingness and responsiveness measures, etcetera.

(4) Authors should explain more detailed the procedure in order to classified adolescents to one of the four categories. Also, authors should indicate the distribution of families by parental styles indicating the number of participants and the percentage, as well as the mean and the standard deviation for each parenting style.

(4) “In the absence of a father and/or mother, the parental style of foster parents (grandparents, uncles and aunts, or other relatives) was assessed. Some adolescents did not answer the items of paternal or maternal style as they did not have persons exerting such roles in their lives.” (see lines 26-29). Authors should indicate details about how many adolescents responded about both parents, about only mother, about only father, or other caregiver instead of non-parents.

(5) Result section constitutes a short of hieroglyphic. Is quite difficult to understand the empirical section and this point should be improved. Authors have in present study repeated measures (before intervention pre-test and after intervention post-test)? In order to test which is the best parenting style, authors should indicate clearly their research design, indicating dependent variables and independent variables. Additionally, authors should conduct multivariate analysis and then univariate in order to test which is the best parenting style (Bono, Arnau, Blanca, & Alarcón, 2016; Gracia, García, & Lila, 2014; O’Brien & Kaiser, 1985)

References

Baumrind, D. (1972). An exploratory study of socialization effects on Black children: Some Black-White comparisons. Child Development, 43, 261-267. doi:10.1111/j.1467-8624.1972.tb01099.x

Bono, R., Arnau, J., Blanca, M. J., & Alarcón, R. (2016). Sphericity estimation bias for repeated measures designs in simulation studies. Behavior Research Methods, 48, 1621-1630. doi:10.3758/s13428-015-0673-1

Calafat, A., García, F., Juan, M., Becoña, E., & Fernández-Hermida, J. R. (2014). Which parenting style is more protective against adolescent substance use? Evidence within the European context. Drug and Alcohol Dependence, 138, 185-192. doi:10.1016/j.drugalcdep.2014.02.705

Chao, R. K. (1994). Beyond parental control and authoritarian parenting style: Understanding Chinese parenting through the cultural notion of training. Child Development, 65, 1111-1119. doi:10.1111/j.1467-8624.1994.tb00806.x

Chao, R. K. (2001). Extending research on the consequences of parenting style for Chinese Americans and European Americans. Child Development, 72, 1832-1843. doi:10.1111/1467-8624.00381

Darling, N., & Steinberg, L. (1993). Parenting style as context: An integrative model. Psychological Bulletin, 113, 487-496. doi:10.1037/0033-2909.113.3.487

Dwairy, M., & Achoui, M. (2006). Introduction to three cross-regional research studies on parenting styles, individuation, and mental health in Arab societies. Journal of Cross-Cultural Psychology, 37, 221-229. doi:10.1177/0022022106286921

Dwairy, M., Achoui, M., Abouserfe, R., & Farah, A. (2006). Parenting styles, individuation, and mental health of Arab adolescents: A third cross-regional research study. Journal of Cross-Cultural Psychology, 37, 262-272. doi:10.1177/0022022106286924

García, F., & Gracia, E. (2009). Is always authoritative the optimum parenting style? Evidence from Spanish families. Adolescence, 44(173), 101-131.

Garcia, O. F., & Serra, E. (2019). Raising children with poor school performance: Parenting styles and short- and long-term consequences for adolescent and adult development. International Journal of Environmental Research and Public Health, 16(1089), 1-24. doi:10.3390/ijerph16071089

Garcia, O. F., Lopez-Fernandez, O., & Serra, E. (2018). Raising Spanish children with an antisocial tendency: Do we know what the optimal parenting style is?. Journal of Interpersonal Violence. doi:10.1177/0886260518818426

Gracia, E., García, F., & Lila, M. (2014). Male police officers' law enforcement preferences in cases of intimate partner violence versus non-intimate interpersonal violence: Do sexist attitudes and empathy matter? Criminal Justice and Behavior, 41, 1195-1213. doi:10.1177/0093854814541655

Maccoby, E. E., & Martin, J. A. (1983). Socialization in the context of the family: Parent-child interaction. In P. H. Mussen (Ed.), Handbook of child psychology (Vol. 4, pp. 1-101). New York: Wiley.

Martínez, I., & García, J. F. (2008). Internalization of values and self-esteem among Brazilian teenagers from authoritative, indulgent, authoritarian, and neglectful homes. Adolescence, 4369), 13-29.

Martínez, I., García, J. F., & Yubero, S. (2007). Parenting styles and adolescents' self-esteem in Brazil. Psychological Reports, 100, 731-745. doi:10.2466/pr0.100.3.731-745

O'Brien, R. G., & Kaiser, M. K. (1985). MANOVA method for analyzing repeated measures designs: An extensive primer. Psychological Bulletin, 97, 316-333. doi:10.1037/0033-2909.97.2.316

Pinquart, M., & Kauser, R. (2018). Do the associations of parenting styles with behavior problems and academic achievement vary by culture? Results from a meta-analysis. Cultural Diversity and Ethnic Minority Psychology, 24, 75-100. doi:10.1037/cdp0000149

Riquelme, M., García, O. F., & Serra, E. (2018). Psychosocial maladjustment in adolescence: Parental socialization, self-esteem, and substance use. Anales de Psicología, 34, 536-544. doi:10.6018/analesps.34.3.315201

Rodrigues, Y., Veiga, F., Fuentes, M. C., & García, F. (2013). Parenting and adolescents' self-esteem: The Portuguese context. Revista de Psicodidáctica, 18, 395-416. doi:10.1387/RevPsicodidact.6842

Steinberg, L., Elmen, J. D., & Mounts, N. S. (1989). Authoritative parenting, psychosocial maturity, and academic-success among adolescents. Child Development, 60, 1424-1436. doi:10.1111/j.1467-8624.1989.tb04014.x

Steinberg, L., Lamborn, S. D., Darling, N., Mounts, N. S., & Dornbusch, S. M. (1994). Over-Time changes in adjustment and competence among adolescents from authoritative, authoritarian, indulgent, and neglectful families. Child Development, 65, 754-770. doi:10.2307/1131416

Valente, J. Y., Cogo-Moreira, H., & Sanchez, Z. M. (2017). Gradient of association between parenting styles and patterns of drug use in adolescence: A latent class analysis. Drug and Alcohol Dependence, 180, 272-278. doi:10.1016/j.drugalcdep.2017.08.015

Author Response

Dear Reviewer 3,

We are submitting a revised version of our manuscript entitled “Role of parenting styles in adolescent substance use cessation: results from a Brazilian longitudinal cohort study”. We have modified the paper in response to the extensive and insightful reviewers comments. In this revised -version, we have addressed the concerns of the editor and the reviewers. Finally, we would like to thank the reviewers for their valuable collaboration in improving the quality of our manuscript.  We listed all changes according to the reviewer 3 suggestions below (marked green color in the revised manuscript):

Response to Reviewer 3 Comments

(1) Parenting styles and adolescent school adjustment.

(1a) Authors should review the theoretical assumptions of Maccoby and Martin’ theoretical framework (1983) about the orthogonality of parental dimensions (warmth and strictness). Also, how authors test whether their measures of parental are non-related or orthogonal? Authors should indicate this point examining the relationship between both parental measures.

The measures of the scale are dimensions of demandingness and responsiveness. We included this suggestion in the third paragraph in the Questionnaire section.

(1b) Authors should review more detailed studies examining parenting styles and developmental outcomes across the globe. Importantly, authoritative parenting is commonly related to optimal adjustment among adolescents, according to findings from studies most of them conducted in middle-class white families from the United States (Darling & Steinberg, 1992; Steinberg, Elmen, & Mounts, 1989; Steinberg, Lamborn, Darling, Mounts, & Dornbusch, 1994). Nevertheless, studies examining parenting styles and adolescent compentence and adjusment in other cultural context indicate that the authoritative parenting styles is not always related to the optimal adjustment (Garcia & Gracia, 2009; Garcia, Lopez-Fernandez, & Serra, 2018; Darling & Steinberg, 1992; Pinquart, 2018). For example, as Darling and Steinberg noted (1992, p 487): “Particularly pressing issues are the variability in the effects of parenting style as a function of the child's cultural background”. And more currently, Pinquart (2018, p. 95) explained that “Associations of parenting style with child outcomes may vary, to some extent, between cultural contexts, such as ethnic groups within countries and different regions of the globe”. On the one hand, some findings from studies with ethnic minority families and dangerous communities seem to suggest that authoritarian parenting may not be as harmful and may even have some protective benefits (Baumrind, 1972; Chao, 1994, 2001; Dwairy & Achoui, 2006; Dwairy, Achoui, Abouserfe, & Farah, 2006). On the other hand, some studies conducted with families from in European and Latin American countries seem to suggest that indulgent parenting is related to equal or even better developmental outcomes than those raised by authoritative parents (Garcia & Gracia, 2009; Garcia & Serra, 2019; Pinquart, 2018).

Thanks for your contribution. We included this suggestion in the Introduction section.

(1c) Authors should review studies in Brazil examining parenting styles and adolescent adjustment and competence. For example, some studies from Brazil (Martínez & García, 2008; Martínez, García, & Yubero, 2007) and Portugal (Rodrigues, Veiga, Fuentes, & García, 2013) seem to indicate that adolescents with indulgent parents obtained equal or even better scores than their peers with authoritative parents, whereas authoritarian and authoritative parenting styles are associated with the lowest scores.

Thanks for your contribution. We included this suggestion in the Introduction section.

 (1d) Authors should add more studies examining parenting styles and drug use outcomes. For example, and interesting study examining parenting styles (indulgent, authoritative, authoritarian, and neglectful) and different indicators of competence an adjustment (including drugs) across five European-countries as Sweden, United Kingdom, Spain, Portugal, Slovenia, and the Czech Republic (Calafat, García, Juan, Becoña, & Fernández-Hermida, 2014) or parenting styles with a multidimensional criteria of drugs (Riquelme, García, & Serra, 2018).Or more currently, in Brazil, Juliana Valente and her colleagues (Valente, Cogo-Moreira, & Sanchez, 2017) have analyzed parenting styles (indulgent, authoritative, authoritarian and neglectful) in order to examine patterns of drug use in adolescence.

Thanks for your contribution. We included this suggestion in the Introduction and Discussion sections.

 (2) Authors should indicate very detailed the family background of participants. Are middle-class families? How author control the key role of socioeconomic status? Information provided by authors (see Table 1) indicated that 60.4% adolescents of adolescents are from families with 1.500,00 Brazilian real (BRL); and those from families with more than 1.500,00 Brazilian real (BRL) represented 39.6%. Why authors have used 1.500,00 Brazilian real (BRL) as criteria? How authors take into account the variable socioeconomic status when they have classified adolescent into one of the four typologies?

We used 1 and a half minimum wage because is once again common and frequent in our parents country. We no take into account the variable socioeconomic status when have classified adolescent.

(3) Measures section is difficult to understand. This point is very important in order to be possible replicate this study. Author should indicate clearly the variable and its specific measure. Authors should provide the items that they used for each measure in a      Table. Importantly, also, authors should indicate alpha values for the subscales. For example, alpha values for demandingness and responsiveness measures, etcetera.

We try to improve this part. It is highlighted in color because other reviewers made similar considerations.

(4) Authors should explain more detailed the procedure in order to classified adolescents to one of the four categories. Also, authors should indicate the distribution of families by parental styles indicating the number of participants and the percentage, as well as the mean and the standard deviation for each parenting style.

The corrections in this regard are marked in red in Materials and Methods section of the manuscript.

(4) “In the absence of a father and/or mother, the parental style of foster parents (grandparents, uncles and aunts, or other relatives) was assessed. Some adolescents did not answer the items of paternal or maternal style as they did not have persons exerting such roles in their lives.” (see lines 26-29). Authors should indicate details about how many adolescents responded about both parents, about only mother, about only father, or other caregiver instead of non-parents.

We included the description of these details in the end of the second paragraph of the Results. 

(5) Result section constitutes a short of hieroglyphic. Is quite difficult to understand the empirical section and this point should be improved. Authors have in present study repeated measures (before intervention pre-test and after intervention post-test)? In order to test which is the best parenting style, authors should indicate clearly their research design, indicating dependent variables and independent variables. Additionally, authors should conduct multivariate analysis and then univariate in order to test which is the best parenting style (Bono, Arnau, Blanca, & Alarcón, 2016; Gracia, García, & Lila, 2014; O’Brien & Kaiser, 1985)

We try to improve this part. It is highlighted in red or blue color because other reviewers made similar considerations.

Round 2

Reviewer 1 Report

The authors have addressed most of the questions mentioned in the review and the revised version of the manuscript has improved. However, there are some question still need to be clarified:

More information of the parental demandingness and responsiveness scale is needed. In page 3 (lines 7-9) the authors write: “The authors of the scale suggest excluding subjects with the same average scores in the parental demandingness and responsiveness subscales”, firs this information should be given after describe the scale. Furthermore, the authors of the scale are not explicated. There is only a reference of use of this scale [reference 21] (Mak & Iacovou, 2019). Information about the development, validation and others uses of this scale is necesary. In page 4 (lines 5-6) the authors write: “The instrument consists of two scales that measure the orthogonal dimensions of demandingness and responsiveness”. Pearson correlation between the two dimensions must be provided in order to show if there are related or unrelated dimensions. The discussion section does not address the incongruence of this study with previous research in Brazil where indulgent parenting shows equal or higher adolescent adjustment than authoritative parenting, one of them published in this special issue (Garcia, F., Serra, E., Garcia, O. F., Martinez, I., & Cruise, E. (2019). The high prevalence of drug use in adolescents’ parents (58% of fathers use substances) must be mentioned as possible explanation of the results, that would situate these families in a context where indulgent style can fail because the family norm is “substance consumption”. This peculiarity of the sample needs to be considered. Furthermore, following recent studies published in this special issue, the sample of this study would be located in a second socialization stage, where authoritative parenting was the optimal style, opposite to the third current stage where indulgent parenting has proved to be optimal (Garcia, F., Serra, E., Garcia, O. F., Martinez, I., & Cruise, E. (2019). In page 10 (lines 14-17) authors are confusing permissive parenting (that group indulgent and negligent) with indulgent parenting. The definition of indulgent parenting is incorrect and does not match with the classification of page 4 (lines 19-23) that defines indulgent parents as “parents scoring low on demandingness and high on responsiveness”. This error of the discussion should be corrected.

References:

Garcia, F., Serra, E., Garcia, O. F., Martinez, I., & Cruise, E. (2019). A third emerging stage for the current digital society? Optimal parenting styles in Spain, the United States, Germany, and Brazil. International journal of environmental research and public health, 16(13), 2333.

Mak, H.W.; Iacovou, M. Dimensions of the Parent-Child Relationship: Effects on Substance Use in Adolescence and Adulthood. Substance use & misuse 2019, 54, 724-736,

Author Response

Dear Reviewer 1,

We have modified the paper in response to the lasted insightful reviewers comments. In this revised -version, we have addressed the concerns of the  reviewers. Finally, we would like to thank the reviewers for their valuable collaboration in improving the quality of our manuscript.  We listed all changes according your suggestions attached (marked red color in the revised manuscript).

Reviewer 2 Report

Thank you authors for the clarifications.

One serious considerations for review--

Why opt for Poisson regression with a binary DV? Poisson models are relevant when your DV is a count measure, i.e., number of instances/events per case/unit of observation. For a binary variable, where outcome is yes/no, the ideal statistical method is logistic regression. The results, as risk ratios, therefore, are confounding. Even with logistic regression, you can assess the odds of change in behavior. Again, with the DV, the change in behavior is not clarified-- is it coded as 1 (yes) if an adolescent has stopped consumption of illicit substances? The presentation of results in Table 3 is also confusing.The Table should present coefficients (betas) and odds ratio only with confidence intervals/z scores and statistical significance, and the total sample size should be mentioned in the Table for each model.

This will alter the discussion and conclusion sections. Paper also requires moderate English editing.

Author Response

Manuscript ID: ijerph-536933
Title:
Role of parenting styles in adolescent substance use cessation: results from a Brazilian longitudinal cohort study
Corresponding author: Mariana Canellas Benchaya

Dear Reviewer 2,

We have modified the paper in response to the lasted insightful reviewers comments. In this revised -version, we have addressed the concerns of the  reviewers. Finally, we would like to thank the reviewers for their valuable collaboration in improving the quality of our manuscript.  We listed all changes according your suggestions bellow.

Reviewer's comments

Thank you authors for the clarifications.

One serious considerations for review--

Why opt for Poisson regression with a binary DV? Poisson models are relevant when your DV is a count measure, i.e., number of instances/events per case/unit of observation. For a binary variable, where outcome is yes/no, the ideal statistical method is logistic regression. The results, as risk ratios, therefore, are confounding. Even with logistic regression, you can assess the odds of change in behavior. Again, with the DV, the change in behavior is not clarified-- is it coded as 1 (yes) if an adolescent has stopped consumption of illicit substances? The presentation of results in Table 3 is also confusing. The Table should present coefficients (betas) and odds ratio only with confidence intervals/z scores and statistical significance, and the total sample size should be mentioned in the Table for each model. This will alter the discussion and conclusion sections.

The relative risk was performed using Poisson regression with robust variance due to the fact that it is a longitudinal study. The odds ratio, estimated by logistic regression could overestimate the effect measure.

Due to the research design was used as a risk force measure the Poisson regression model with robust variance, so the odds ratio was not showed.

Paper also requires moderate English editing.

We send the English certificate of correction by American Journal Experts.

Reviewer 3 Report

Authors have made some important changes in order to improve the quality of their study. Nevertheless, some important points still remain unclearly.

* Authors should clearly indicate how they categorize thefamilies as indigent, authoritative, authoritarian or neglectful. Authors should offer more details. How many participants were excluded of the study according to the criteria? Why authors not considered two participants with the same score in one or both parental measures (responsiveness and demandingness)? For each participant, authors calculated a score in responsiveness and demandingness by the mean of father and mother?Parental typologies (indulgent, authoritative, authoritarian or neglectful) are obtained by the mean or by the general median (for father, for mother, or for both)?

* Authors should review label “negligent”. Where said “negligent” should said “neglectful”

* Authors should review the style of statistics. Statistics should be in italic. Check it out in all manuscript.

* Authors should define clearly the socioeconomic status of families. In this sense, it seems that authors define families of preset study as low-income working-class families instead of middle-class families. As authors pointed out, “60.4% were in an household with an income lower 1 and a half minimum wage (Brazilian real)” (see lines 46-47).

* Authors should discuss their present results about the benefits of authoritative parenting considering the socioeconomic status of families (low-income working-class families), and according to current results about the optimal parenting stages (e.g., Garcia, Serra, Garcia, Martinez, & Cruise, 2019). In this sense, specifically low-income working-class families authors found benefits related to authoritative parenting style.

References

Garcia, F., Serra, E., Garcia, O. F., Martinez, I., & Cruise, E. (2019). A third emerging stage for the current digital society? Optimal parenting styles in Spain, the United States, Germany, and Brazil. International journal of environmental research and publichealth16(13), 2333. doi:10.3390/ijerph16132333

Author Response

Dear Reviewer 3,

We have modified the paper in response to the insightful reviewers comments. In this revised -version, we have addressed the concerns of the reviewers. Finally, we would like to thank the reviewers for their valuable collaboration in improving the quality of our manuscript. We listed all changes according to the reviewer your suggestions below (marked green color in the revised manuscript):

Reviewer's comments

Authors have made some important changes in order to improve the quality of their study. Nevertheless, some important points still remain unclearly.

* Authors should clearly indicate how they categorize the families as indigent, authoritative, authoritarian or neglectful. Authors should offer more details. How many participants were excluded of the study according to the criteria? Why authors not considered two participants with the same score in one or both parental measures (responsiveness and demandingness)? For each participant, authors calculated a score in responsiveness and demandingness by the mean of father and mother? Parental typologies (indulgent, authoritative, authoritarian or neglectful) are obtained by the mean or by the general median (for father, for mother, or for both)?

Thanks for your questions. More details about how we categorize the families and how many participants were excluded are showed in the Questionnaire Section.

We clarify about parental typologies in the Questionnaire Section. Responsiveness and demandingness parental scores were obtained by the median of the sample, separately for the father and mother scores.

* Authors should review label “negligent”. Where said “negligent” should said “neglectful”

Thanks for your correction, we did.

* Authors should review the style of statistics. Statistics should be in italic. Check it out in all manuscript.

Thanks for your correction, we did.

* Authors should define clearly the socioeconomic status of families. In this sense, it seems that authors define families of preset study as low-income working-class families instead of middle-class families. As authors pointed out, “60.4% were in an household with an income lower 1 and a half minimum wage (Brazilian real)” (see lines 46-47).

The Ligue 132 service is a free, a telephone counseling service. In general, the population of Ligue 132 was low-income.

* Authors should discuss their present results about the benefits of authoritative parenting considering the socioeconomic status of families (low-income working-class families), and according to current results about the optimal parenting stages (e.g., Garcia, Serra, Garcia, Martinez, & Cruise, 2019). In this sense, specifically low-income working-class families authors found benefits related to authoritative parenting style.

We agree and believe that discussing the issue of socioeconomic status is critical. Therefore, we have included a new paragraph in the discussion.

Round 3

Reviewer 1 Report

The authors have addressed the most important points pointed out in the last review and the manuscript is ready to be published. However, there is still an important mistake that must be corrected in relation to the measure of demandingness and responsiveness.

In page 3 lines 49-50 it's written: “The instrument consists of two scales that measure the orthogonal dimensions of demandingness and responsiveness”. But just below the authors mention that the two dimensions are correlated. Therefore, the two dimensions of this scale are not orthogonal, since orthogonal means unrelated.

The authors need to clarify this by adding a sentence saying that: ·In this specific instrument the dimensions are not orthogonal, although theoretically they are. Just how it happens in other well know measures of parenting (e.g. Steinberg, Lamborn, Darling, Mounts, Dornbusch) (see Ker & Stattin, 2000; Stattin, & Kerr, 2000, or Martínez, García, Fuentes, et al., 2019 in this Special Issue for a better understanding of this concept and his measurement)

Steinberg, L.; Lamborn, S.D.; Darling, N.; Mounts, N.S.; Dornbusch, S.M. Over-Time Changes in Adjustment and Competence among Adolescents from Authoritative, Authoritarian, Indulgent, and Neglectful Families. Child Dev. 1994, 65, 754–770, doi:10.2307/1131416.

Kerr, M.; Stattin, H. What Parents Know, how they Know it, and several Forms of Adolescent Adjustment: Further Support for a Reinterpretation of Monitoring. Dev. Psychol. 2000, 36, 366–380, doi:10.1037/0012-1649.36.3.366.

Martinez, I.; Garcia, F.; Fuentes, M.C.; Veiga, F.; Garcia, O.F.; Rodrigues, Y.; Cruise, E.; Serra, E. Researching Parental Socialization Styles Across Three Cultural Contexts: Scale ESPA29 Bi-Dimensional Validity in Spain, Portugal, and Brazil. Int. J. Environ. Res. Public Health 2019, 16, 197, doi:10.3390/ijerph16020197.

Stattin, H.; Kerr, M. Parental Monitoring: A Reinterpretation. Child Dev. 2000, 71, 1072–1085, doi:10.1111/1467-8624.00210.

Author Response

Dear Reviewer 1,

We have modified the paper in response to the lasted insightful reviewers comments. In this revised -version, we have addressed the concerns of the  reviewers. Finally, we would like to thank the reviewers for their valuable collaboration in improving the quality of our manuscript.  We listed all changes according your suggestions attached (marked red color in the revised manuscript):

We included the correction in Questionnaire Section, lines 2-5, page 4.